microbiology, plant science, ecology

pea aphid (*Acyrthosiphon pisum*), *Medicago truncatula*, symbiosis, rhizobium, nitrogen fixation, plant defence

**Author for correspondence:**
Marylène Poirié
e-mail: marylene.poirie@univ-cotedazur.fr

†These authors are co-last author.

# Aphid infestation differently affects the defences of nitrate-fed and nitrogen-fixing *Medicago truncatula* and alters symbiotic nitrogen fixation

Gaurav Pandharikar[1], Jean-Luc Gatti[1], Jean-Christophe Simon[2], Pierre Frendo[1,†] and Marylène Poirié[1,†]

[1]Université Côte d'Azur, INRAE, CNRS, ISA, France
[2]INRAE, UMR IGEPP, Domaine de la Motte, Le Rheu, France

GP, 0000-0002-4318-5549; J-LG, 0000-0001-7683-718X; J-CS, 0000-0003-0620-5835;
PF, 0000-0002-4578-3366; MP, 0000-0002-3349-6925

Legumes can meet their nitrogen requirements through root nodule symbiosis, which could also trigger plant systemic resistance against pests. The pea aphid *Acyrthosiphon pisum*, a legume pest, can harbour different facultative symbionts (FS) influencing various traits of their hosts. It is therefore worth determining if and how the symbionts of the plant and the aphid modulate their interaction. We used different pea aphid lines without FS or with a single one (*Hamiltonella defensa*, *Regiella insecticola*, *Serratia symbiotica*) to infest *Medicago truncatula* plants inoculated with *Sinorhizobium meliloti* (symbiotic nitrogen fixation, SNF) or supplemented with nitrate (non-inoculated, NI). The growth of SNF and NI plants was reduced by aphid infestation, while aphid weight (but not survival) was lowered on SNF compared to NI plants. Aphids strongly affected the plant nitrogen fixation depending on their symbiotic status, suggesting indirect relationships between aphid- and plant-associated microbes. Finally, all aphid lines triggered expression of *Pathogenesis-Related Protein 1* (*PR1*) and *Proteinase Inhibitor* (*PI*), respective markers for salicylic and jasmonic pathways, in SNF plants, compared to only *PR1* in NI plants. We demonstrate that the plant symbiotic status influences plant–aphid interactions while that of the aphid can modulate the amplitude of the plant's defence response.

## 1. Introduction

Symbiosis, the intimate relationship between two or more living organisms, is an evolutionary force shaping life on our planet. Well-known examples are the extended phenotypes that bacterial symbionts confer to plants or animals. Legumes (Fabaceae) are unique by the symbiosis they can establish with nitrogen-fixing soil bacteria, Rhizobia [1,2], which can reduce atmospheric nitrogen ($N_2$) to ammonia usable by plants. This biological symbiotic nitrogen fixation (SNF) occurs in root nodules, specialized plant organs induced by the bacterium [3]. SNF improves the productivity of leguminous crops by increasing soil fertility, therefore benefiting associated crops in intercropping [4]. The presence of rhizobacteria also improves the plant response to pathogens and herbivores thanks to several mechanisms such as nutrient competition and the triggering of induced systemic resistance (ISR) [5,6]. ISR resembles pathogen-induced systemic acquired resistance (SAR), both of them increasing the resistance of uninfected plant parts to a wide range of pathogens through plant hormones induction, salicylic acid (SA), jasmonic acid (JA) and ethylene (ET), the major plant defence signalling pathways against pathogens and insects [7]. By producing SA on the root surface, several rhizobacteria trigger the SA-dependent

pathway while others activate an SA-independent pathway [7]. However, the effectiveness of ISR-triggered plant defence depends on genetic and environmental factors. Overall, microbe–plant–insect interaction, called 'three-way interaction', is an expanding research area [8]. Aphids are a serious pest of many crops, ornamental plants, or forest trees. Of the 4000 known aphid species, 450 thrive on crops and about 200 cause serious damage by sucking the phloem sap up, reducing plant growth and, most importantly, transmitting plant viruses [9,10]. Most aphids live in obligate symbiosis—over 150 Myr old—with the gamma-proteobacterium *Buchnera aphidicola*, which provides essential amino acids absent from the phloem sap [11,12]. *Buchnera* bacteria are housed in specialized cells, the bacteriocytes and they are transmitted vertically [13]. In addition, aphids can host one or a few heritable facultative symbionts (FS), nine of which can be found in pea aphid populations [14,15]. These bacteria are not essential for aphid survival and reproduction, and can even be costly [16,17], but they provide the host with extended phenotypes including resistance to parasitoid wasps and pathogenic fungi, and heat stress tolerance [18]. The pea aphid (*Acyrthosiphon pisum*) feeds on legumes and forms a complex of at least 15 plant-adapted biotypes, each of them specialized on different legume hosts [19,20]. Interestingly, some biotypes are specifically associated with a facultative symbiont [21,22], suggesting that symbionts can increase the host performance on specific plants [15,18,23].

Since the economic impact of aphids is linked to successful colonization and establishment on host plants, understanding how the associated FS influence them is essential for the management of these pests. Surprisingly, only a few studies on aphid–plant–microbe interactions specified the symbiotic state of the legumes used, usually fava beans, the universal plant host for all pea aphid biotypes. However, the nitrogen-fixing symbiosis of the plant could in turn affect the aphid phenotypes. Indeed, the plant-associated symbionts can affect the performance of some insects in terms of feeding efficiency, metabolism and ability to manipulate the physiology of the plant [5]. Besides, insect endosymbionts can directly affect the performance of their host by decreasing its reproduction and immunity depending on the nutritional status of the plant [24]. They can also indirectly interfere with plant signal transduction pathways by repressing or neutralizing defence-related responses or altering plant metabolism [25]. Although solid fundamental knowledge is available on plant–microbe and insect–microbe interactions, the indirect relationships between the four protagonists still need to be deepened. Therefore, since specific aphid lines can be produced by elimination/injection of FS, we studied the FS potential influence on legume–aphid interactions, considering the plant in symbiosis (SNF) or supplemented with nitrate (non-inoculated, NI). We used *Medicago truncatula*, a legume–rhizobia symbiotic model, and *A. pisum* (pea aphid) lines of identical genetic background (YR2 clone) deprived of FS or hosting one of the most common FS in the field (*Hamiltonella defensa*, *Regiella insecticola*, *Serratia symbiotica*) [18,26]. Our objective was to test for the influence of the plant SNF on the development and growth of aphids depending on the facultative hosted symbiont, and of each facultative symbiont on the SNF and NI plants. We showed that biological nitrogen fixation reduces the aphid fitness regardless of aphid lines compared to nitrate feeding (NI) conditions. Infection with most aphid lines significantly reduced the efficiency of

nitrogen fixation in plants by affecting the function of root nodules estimated by chemical assay and expression of specific root nodules genes (*leghemoglobin* and *cysteine protease 6*). Finally, all aphid lines triggered the expression in plant shoots of *Pathogenesis-Related Protein 1* (*PR1*), a well-defined gene marker for salicylic defence pathway [27], and *Proteinase Inhibitor* (*PI*), a marker for jasmonate defence pathway [28], in SNF plants, while only *PR1* expression was triggered in NI plants. Overall, we demonstrate that the outcome of plant–aphid interactions is influenced by the plant symbiotic status and modulated by the aphid-hosted symbiont.

## 2. Material and methods

### (a) Plant material and growth conditions
*Medicago truncatula* Jemalong A17 is susceptible to different lines of the pea aphid [29]. The seeds were prepared as previously described [30,31] (see electronic supplementary material, figure S1 for details). After germination, six pots, each containing six plants, were inoculated with *Sinorhizobium meliloti* 2011 strain (SNF plants) and six pots, each containing six plants, were supplemented with 5 mM KNO$_3$ solution (NI plants) (details can be found in electronic supplementary material, figure S1) [32,33].

### (b) Aphids rearing and infestation
Five *A. pisum* lines of YR2 genetic background were used [26]. The YR2 clone was collected on red clover (*Trifolium pratense*) and naturally hosts an *R. insecticola* strain ($Ri^{YR2}$), therefore being called here YR2-*Ri*(n). All YR2 lines differ only in their facultative symbiont(s): YR2-Amp, devoid of facultative symbiont, derives from YR2-*Ri*(n) by ampicillin treatment [26,34]. YR2-Amp was used to produce the lines YR2-*Ri*(a) ((a) for artificial), YR2-*Hd* and YR2-*Ss* by injection of, respectively, *R. insecticola* from the T3-8V1 clone ($Ri^{T3-8V1}$ strain), *H. defensa* from the L1-22 clone and *S. symbiotica* from the P136 clone [26]. The YR2-*Ss* line is co-infected with *Rickettsiella viridis* due to its presence in the P136 donor clone. All these lines were stable and reared in aerated cages on four-week-old *Vicia faba* plants at 20°C with a 16/8 h light/dark photoperiod. The symbiotype was controlled by PCR at different times as previously described [35].

### (c) Experimental design and analysis of the biological material
Plants were infested by aphids one-week post-inoculation with *S. meliloti* or nitrate supplementation, a time at which the dry weight of the two plant types was almost identical (electronic supplementary material, figure S2). Each of the five out of six groups of plants was infested with one of the YR2 aphid lines (YR2-Amp, -*Ri*(n), -*Ri*(a), -*Hd* and -*Ss*), and the sixth, left uninfested, served as a control. Ten synchronized L1 nymphs per pot were used for infestation [36] (see electronic supplementary material, figure S1).

### (d) Analyses of aphid fitness and aphid effect on plants
Aphid survival was assessed daily and the average weight of surviving aphids estimated on day 12, just before adult aphids started to reproduce, enough time to establish the plant's nitrogen-fixing symbiosis and its defence response to aphids. The effect of the five aphid lines on SNF and NI plants was estimated by weighing the plant shoots after removal of the aphids. For dry weight, the shoots were placed in a drying oven at 80°C for 3 days and weighed on a precision balance (PA214; OHAUS Corp, accuracy ±0.1 mg). The six individual plants from pots for each plant

condition were weighted individually; means were calculated from three separated experiments.

## (e) Nitrogen fixation assay

The nitrogen fixation assay was done on the roots of SNF plants immediately after collection. Nitrogen-fixing ability of the nodules was estimated indirectly by the reduction of acetylene to ethylene by the nitrogenase (acetylene reduction assay: ARA). The nodulated roots were incubated at 28°C for 1 h in rubber-capped glass bottles containing acetylene. Gas conversion was measured by gas chromatography (6890 N GC network system, Agilent). After ARA measurement, the nodules were separated from the roots, counted and weighed. The ARA values were expressed in nmol of ethylene $\times h^{-1} \times mg$ of nodule$^{-1}$ [ARA/(h × mg nod)] and in nmol of ethylene $\times h^{-1} \times plant^{-1}$ [ARA/plant]. The six individual plants per pot were divided into two technical repeats for the three separate biological experiments.

## (f) Gene expression analysis by quantitative RT-PCR

The six SNF and NI plant shoots and the SNF plant nodules were collected immediately after the aphid removal, pooled and frozen in liquid nitrogen. The plant material was then powdered in liquid nitrogen and total RNAs were isolated using RNAzol (Sigma), spectrometrically quantified (NanoDrop; Thermo Scientific), and their purity assessed on Bioanalyser chips (Agilent) and on 1.5% agarose gel. DNA digestion (RQ1 RNAse-free DNAse) and reverse transcription (GoScript Reverse Transcription) were performed as described by the manufacturer (Promega). The quantitative PCR was performed using the qPCR Master Mix plus CXR (qPCR kit; Promega), with 0.125 µl of cDNA template and each set of specific primers (more details in electronic supplementary material, table S1). Defence-related genes used were *PR1* (pathogenesis-related 1; *Medtr2g435490*) (NCBI; XP_013463163.1) and *Medtr4g032865*, coding for proteinase inhibitor PSI-1.2, a potato type II proteinase inhibitor family protein, thereafter named proteinase inhibitor *PI* (NCBI; KEH29269.1). For the root/nodule function, we used the leghemoglobin-1 gene *MtLb1* (*Medtr5g066070*) (NCBI; XP_003615280.1) since it participates in the protection of the nitrogenase from oxygen denaturation and provides oxygen for bacterial respiration [37] and *Medtr4g079800* encoding the senescence-specific cysteine protease SAG39 also named Cp6 (NCBI; XP_003607574.1) [38]. Real-time RT-qPCR was performed and analysed as indicated in electronic supplementary material, table S1. Cycle threshold values (Ct) were normalized using two housekeeping genes *MtC27* and *a38*. Calculations were done with the RqPCRBase package [39] using RStudio v. 1.1.453 (https://www.rstudio.com). Results were from four independent biological repeats with three technical repetitions per experiment.

## (g) Statistical analysis

All experimental data were expressed as mean ± s.e. To test for an effect of the plant treatment (SNF and NI) on the survival and weight of aphid lines, data were analysed using a two-way ANOVA. This allowed testing the effect of two independent variables (plant condition (SNF or NI); different aphid lines (Amp, -*Ri*(n), -*Ri*(a), -*Hd* and -*Ss*)) on the weight and survival of aphids at 12 dpi. Differences between the aphid lines were tested using the Sidak's multiple comparison test. Data generated on the plant dry weight, the nitrogen fixation assay per plant or per mg of nodule, the number of root nodules per plant and the weight of the nodules per plant were analysed using a one-way ANOVA. Results from these experiments on SNF and/or NI plants were compared independently based on the treatment (one-way ANOVA). Then, Tukey multiple comparison tests were performed in independent treatments to identify possible statistical differences between the aphid lines.

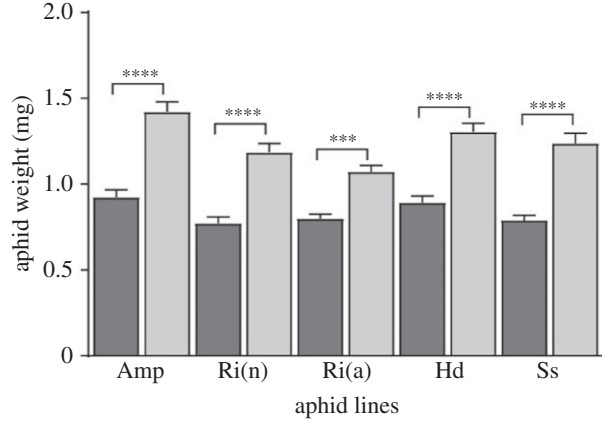

**Figure 1.** Weight of pea aphids from the different lines on *M. truncatula* SNF and NI plants. The mean individual weight of surviving adult aphids of the different lines on SNF (dark grey) and NI (light grey) plants; mean ± s.e., $n = 18$. Statistically significant differences: ***$p \leq 0.001$; ****$p \leq 0.0001$.

All experimental data, except expression analysis results, were analysed using Prism v. 7 (GraphPad software, USA). Data are available in electronic supplementary material, tables S2–S10.

## 3. Results

### (a) Effect of symbiotic nitrogen fixation and non-inoculated plants on the survival and weight of pea aphid lines

To ascertain that *A. pisum* YR2 lines could develop on *M. truncatula* SNF and NI plants, the number and weight of surviving adult aphids, two fitness proxies [40], were estimated 12 days post-infestation (dpi). About 90% of aphids reached adulthood on both types of plants, with a significant difference between the lines [26,34] but not between the two plant conditions for each line (electronic supplementary material, figure S2 and table S2). By contrast, the average weight of surviving aphids was significantly higher (at least 40%) on NI plants than on SNF, regardless of the line (figure 1).

### (b) Effects of aphid infestation on the biomass of *M. truncatula* plant shoots

At the beginning of the experiment, the shoots of SNF and NI control plants had a similar dry weight (electronic supplementary material, figure S3) while 12 days later, it was about twice as high for NI control plants than for SNF (figure 2, control) [33]. Infestation of SNF plants with YR2-Amp, -*Ri*(a) and -*Ss* lines significantly reduced their dry weight compared to the control (about 25%) (figure 2a), unlike that with YR2-*Ri*(n) and -*Hd* (10%) which was not significant. The dry weight of NI plants was considerably reduced after infestation with YR2-Amp, -*Ri*(n), -*Ri*(a) and -*Hd* (also around 25%), but not with YR2-*Ss* (15% reduction, non-significant) (figure 2b). The reduction in growth of aphid-infested plants therefore occurs regardless of the nitrogen nutrition mode but its amplitude depends on the FS hosted.

### (c) Effect of aphid lines on the nitrogen fixation of symbiotic nitrogen fixation plants

The effect of pea aphid lines on the biological nitrogen fixation was first evaluated by counting and weighing the nodules

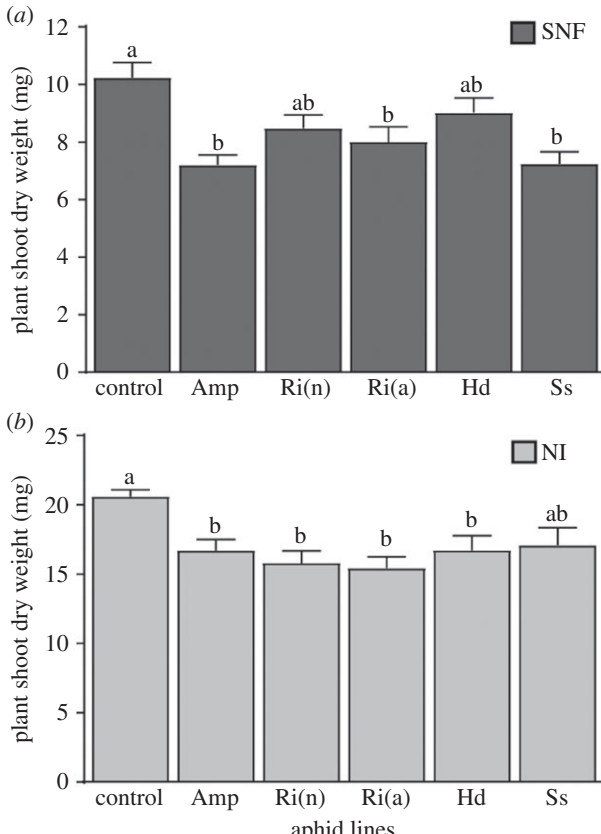

**Figure 2.** Effect of pea aphid lines infestation on the dry weight of *M. truncatula* plant shoots. Dry weight of *M. truncatula* SNF (*a*) and NI (*b*) plant shoots after 12 days of infestation with the different pea aphid lines compared to their respective non-infested control plants; mean ± s.e., *n* = 3. Statistical differences between the means are indicated by different letters ($p \leq 0.05$).

(figure 3) as they are macroscopic markers for the establishment of SNF in plants [41]. The number of root nodules per plant was significantly lower in aphid-infested plants than in non-infested control plants, except for YR2-*Hd* (figure 3*a*). The highest effect was for YR2-*Ss* and YR2-*Amp* with a 50% reduction in the number of nodules. The total weight of nodules per plant was also reduced in infested plants, but significantly only for the YR2-*Ss* and -*Amp* lines (figure 3*b*).

We then measured the nitrogen-fixing activity of the root nodules using the acetylene reduction assay (ARA). The plants infested with YR2-*Amp*, -*Hd* and -*Ss* showed a significant reduction in the ARA expressed per mg of nodule compared to the control while this reduction was not significant with YR2-*Ri*(n) and YR2-*Ri*(a) (figure 3*c*). The ARA per plant gave a similar result, except for the reduction with YR2-*Ri*(a) which was significant here (figure 3*d*). Nodules nitrogen fixation efficiency was therefore affected by aphid infestation. To further investigate this effect on the biological function of nodules, we estimated the expression rate of two specific genes by qRT-PCR: (i) the leghemoglobin gene *Mtlb1*, whose expression correlates with optimal nitrogen fixation [42], and (ii) the *MtCP6 cysteine protease* (*Cp6*), a gene expressed during both developmental and stress-induced nodule senescence [38]. Only the infestation with YR2-*Ri*(a) and YR2-*Hd* reduced significantly the expression of *Mtlb1* in SNF plants (figure 4*e*; electronic supplementary material, table S10). By contrast, the expression of *MtCp6* was increased 5–23-fold in infested SNF plants, except with

YR2-*Ri*(n) for which there was no significant change in expression (figure 4*f*; electronic supplementary material, table S10). Overall, data suggest an early induction of nodule senescence in infested SNF plants and therefore a decrease in metabolic efficiency.

## (d) Expression of jasmonic acid and salicylic acid plant defence pathways

Aphid feeding is known to induce expression of gene markers of the SA pathway, such as the *PR1* gene [43–45]. It has also been speculated that activation of the SA-signalling pathways counteracts the activation of defence responses related to JA. Here, we have also analysed the expression of *PI*, a marker of the JA pathway [28]. The expression of *PR1* and *PI* was similar in the shoots of non-infested SNF and NI control plants (electronic supplementary material, figure S4), suggesting that the plant nutrition mode does not affect the basal defences of the plants. In infested plants, *PR1* expression was significantly increased (from 9-fold in SNF plants with YR2-*Amp* to 50-fold in NI plants with YR2-*Ri*(a)), regardless of the nutrition mode, except for YR2-*Hd* on NI plants (figure 4*a*), suggesting an activation of the SA pathway. The most striking result, however, was the contrasting level of expression of *PI* upon aphid infestation: SNF plants showed a significant five- to eight-fold induction in shoots (figure 4*b*). By contrast, a lower increase in *PI* expression was observed in roots of infested SNF plants, only significant for YR2-*Hd* (electronic supplementary material, figure S5), and none in roots of NI plants. This provides evidence for a differential regulation of plant defence mechanisms by aphids according to the presence of rhizobium and to a lesser extent of the aphid FS.

## 4. Discussion

### (a) Aphid weight is mainly affected by the mode of nitrogen nutrition of the plant

Facultative aphid symbionts (FS) can shape the phenotype of their hosts, including their interactions with host plant [46]. For instance, their presence can alter the systemic release of volatiles by plants after aphid attack [47], increasing their fitness. Moreover, *A. pisum* biotypes adapted to a given legume are strongly associated with a particular FS [18,23] whose removal can affect their fitness on this plant [23]. Likewise, we believe that the presence of some FS may modulate the plant–aphid interaction. The pea aphid clone used here, YR2, belongs to the clover biotype [26] and naturally hosts *R. insecticola* (*Ri*^YR2 strain). This clone is undoubtedly not the best adapted to *Medicago*, but we showed here that all the lines were able to feed and develop on *M. truncatula* plants as indicated by the low aphid mortality and as previously reported [48]. Adult females of all lines were also able to reproduce during at least two weeks on SNF or NI plants and preliminary data suggest a higher offspring number on NI plants than on SNF, except for YR2-*Ri*(a), and therefore a positive correlation between weight and fecundity (G.P. 2019, personal observation). In the field, pea aphids infected with *Hd* or *Ri* are commonly found on *Medicago* plants, unlike those infected with *Ss* [49]. However, despite variation in symbiotic status, the different YR2 lines

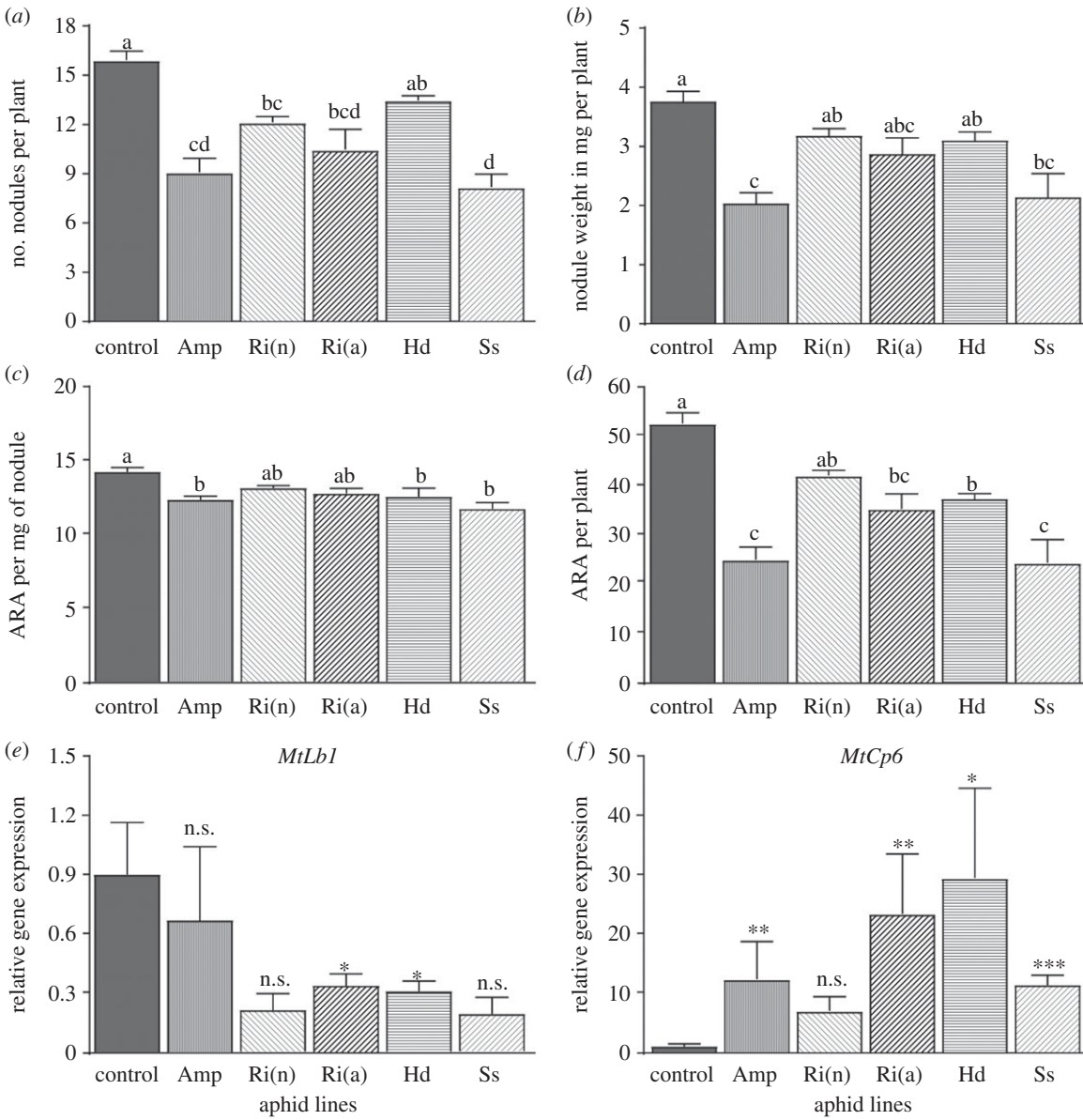

**Figure 3.** Effects of pea aphid lines infestation on biological nitrogen fixation. (*a,b*) Number (*a*) and weight (*b*) of root nodules from SNF plants infested or not (control) with the different YR2 pea aphid lines for 12 days. (*c,d*) Mean acetylene reduction activity expressed per milligram of nodule (nmol ethylene $\times$ h$^{-1}$ $\times$ mg nodule$^{-1}$) (*c*) and per plant (nmol ethylene $\times$ h$^{-1}$ $\times$ plant$^{-1}$) (*d*) after infestation with pea aphids compared to non-infested plants (control). (*e,f*) Relative expression (qPCR) of (*e*) *Mtlb1* and (*f*) *MtCp6* in the nodules of plants infested by pea aphid lines compared to those of non-infested plants. In (*a–d*): mean $\pm$ s.e., $n = 3$; statistical differences among means are indicated by different letters ($p \leq 0.05$). In (*e,f*): mean $\pm$ s.e., $n = 4$. Statistical differences: n.s., not significant ($p \geq 0.05$); *$p \leq 0.05$; **$p \leq 0.01$; ***$p \leq 0.001$ (see also electronic supplementary material, table S11).

had a fairly similar individual weight, the main effect being due to the nutritional state of the plant with a lower weight on SNF plants, which were also less developed at the end of the experiments (see below). This suggests either a lower nutritional quality of their sap for aphids or lower acceptance of SNF plants by aphids (lower feeding uptake).

## (b) Aphid infestation affects plants according to their nitrogen nutrition mode and the facultative symbiont hosted

The plant dry weight increased twice as fast for NI than SNF plants during the experiment. This growth retardation of SNF plants certainly results from the higher energy cost of establishing a symbiotic fixation of nitrogen compared to nitrate supply, possibly combined with a limited effectiveness of the symbiotic association between *M. truncatula* A17 and

Sm2011 [33,50]. This association with the Sm2011 strain could therefore accentuate the response of the nodulated plants to the infestation of aphids. However, whatever the nitrogen source of the plant, aphid infestation reduced the dry weight of the shoots, with an amplitude dependent on the FS hosted. *Hd* and *Ri* still have pathogenic traits, unlike *Ss*, and *Ri*$^{YR2}$ (in YR2-Ri(n)) and *Ri*$^{T3-8V1}$ (in YR2-Ri(a)) are different bacterial strains [14]. FS species/strains were therefore expected to affect their aphid host and the host plant differentially. Aphids inject saliva that contains proteins and metabolites which are thought to facilitate feeding and modulate plant physiology [45] including defence mechanisms [51]. Aphid saliva may also contain proteins derived from symbionts, such as the chaperone GroEL, which has been showed to elicit plant defence [52]. As aphids feed on the plant, the presence of FS can also influence competition for metabolites. Little is known about the effect of FS on the metabolic needs of aphids and their salivary components

Proc. R. Soc. B 287: 20201493

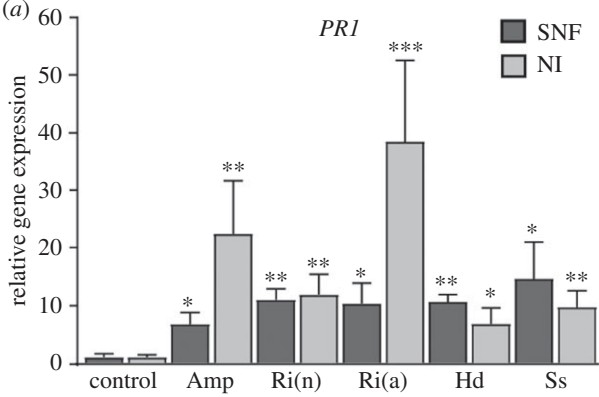

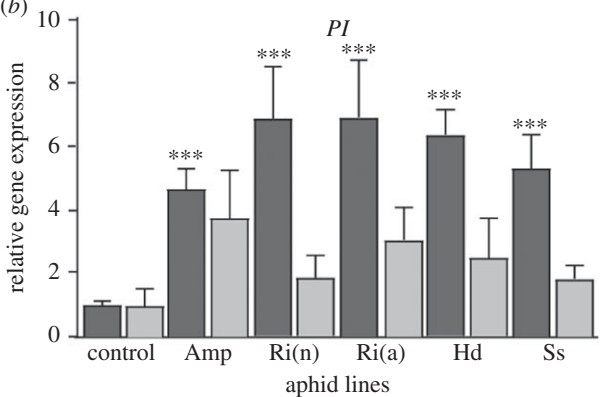

**Figure 4.** Expression level of defence-related genes. (*a,b*) Expression of *PR1* (*a*) and *PI* (*b*) estimated by qPCR in shoots of SNF and NI plants 12 days after infestation by the different YR2 aphid lines. Non-infested plants serve as the basal level for comparison (control). Mean ± s.e., *n* = 4. Statistical differences (*t*-test): *$p \leq 0.05$; **$p \leq 0.01$; ***$p \leq 0.001$.

[53] or the role of the compounds they inject, but the observed differences here deserve further attention.

## (c) Aphid infestation affects the biological nitrogen fixation of the plant

In addition to reducing the growth of SNF plants, aphid infestation resulted in three major congruent effects related to nitrogen fixation but depending the aphid line: the reduction in the number of root nodules, the decrease in efficiency of nitrogen fixation per nodule (ARA) and the repression of the *leghemoglobin* gene (*Mtlb1*), an indicator of optimal nitrogen fixation. Taken together, these effects result in reduced efficiency of nitrogen fixation in plants, which may partly explain the lower growth of infested SNF plants. The reduced number of nodules suggests a blockage of nodulation due to aphid infestation either during the process of rhizobium infection and/or of the formation of nodules. This fits with the increased expression of *Cp6* which suggests an earlier induction of nodule senescence in infested plants [54,55], possibly explaining part of the reduction in nitrogen-fixing activity. The presence of certain FS restricts the aphid ability to feed on SNF plants, therefore lowering the induced reduction of nitrogen fixation. *Ri* and *Hd*, notably, might somehow offset the effect of aphid infestation.

Studies considering plant–pathogen interactions as a factor limiting the establishment of a nitrogen-fixing symbiosis are still scarce [56]. Co-inoculation of *Rhizoctonia solani* or *Sclerotinia sclerotiorum* with rhizobia reduces the number of nodules—also observed in the infection of *M. truncatula* with

the leaf pathogen *Pseudomonas syringae*—and dry matter of the roots [57,58]. By contrast, infestation of *Alnus viridus* with the specific aphid *Prociphilus tessallatus*, which failed to establish feeding colonies, increased nitrogen-fixing activity and plant growth [59]. The question of how aphids impact root nodules and how FS influence this effect remains to be elucidated. One hypothesis would be the occurrence of a trade-off in the plant, the aphid infestation leading to the inhibition of the costly formation of nodules to compensate for the uptake of metabolites from the plant sap. This would in turn decrease the availability of nitrogen-containing metabolites, such as amino acids, for aphids, which can decrease their appetence for the plant or their fitness. Indeed, it has been shown that the Buchnera density was positively correlated with the aphid dietary nitrogen levels [60]. Conversely, *S. symbiotica* number increased in aphids reared on a low nitrogen diet, demonstrating possibly distinct regulatory mechanism or nutritional needs between symbionts in the same insect host. Moreover, the composition of the phloem sap may differ between nitrate-fed and nitrogen-fixing plants which could modulate the result of the association by reducing aphid growth on SNF plants. Analysing the sap composition of SNF and NI plants with or without aphids will be crucial to test this hypothesis. Another way, but not exclusive, by which aphid infestation may affect the function of nodules is through the activation of plant defence pathways (see below).

## (d) Aphid infestation affects the defences of symbiotic nitrogen fixation and non-inoculated plants differentially

Aphids may induce three main plant defence signalling networks: SA, JA and ET [61,62]. They elicit them through cell damage, the production of reactive oxygen species during penetration of the stylet and the recognition of their salivary components by the plant [59]. Crosstalk between the SA and JA pathways plays a crucial role in triggering defences against agressors [63]. JA production is generally linked to the defences against necrotrophic microbes and chewing herbivores and that of SA to those against biotrophic pathogens [64,65]. It has been shown that aphid feeding could increase the transcription of several PR genes and others associated with the (SA)-dependent response, resulting in an increase in enzymatic activities such as those of peroxidases and chitinases [45]. For example, feeding of the green peach aphid *Myzus persicae* on *Arabidopsis thaliana* and *Solanum tuberosum* L. respectively induced a 10-fold increase in the transcription of *PR1* and a production of PR1 transcripts gradually increasing over the feeding time of aphids [43,66]. The feeding of pea aphids on *M. truncatula* resulted in a two to threefold increase in *PR5* expression in the first 3 days after infestation [29]. Here, *PR1* expression was strongly upregulated in infested plants under almost all our conditions, confirming the activation of SA-dependent defences, regardless of the nitrogen nutrition of the plants. Activation of the SA pathway may also be a general mechanism of antibiosis or aphid repulsion [64,65], but its level being similar in SNF and NI plants, it cannot explain their differential effect on the aphid weight.

The JA pathway was differentially activated between infested SNF and NI plants, the expression of the (*PI*) marker gene being significantly upregulated in SNF plants only. This is in agreement with the results of Gao *et al.* [29], who observed

no increase in *PI* expression in non-inoculated *M. truncatula* infested with the pea aphid. Several thousands of genes are up- or downregulated in root tissues when establishing the *Sinorhizobium* symbiosis, including genes from the JA pathway [67]. Moreover, certain strains of *S. meliloti*, including Sm2011, induce *M. truncatula* defence responses similar to the pathogenic *P. syringae* strain DC3000, although this may be transient [58]. The JA pathway could thus be sensitized by the presence of *S. meliloti*, aphid feeding then being sufficient to trigger it at a significant level. JA has many roles in plants apart from immune defence such as promoting growth and development, including the formation of leaves and roots, and controlling reproduction and senescence [68]. JA, alone or by its ratio to ethylene, has also been linked to blockage of nodulation [69], suggesting that it could mediate this aphid-induced effect. Yet, in our experience, the *PI* expression in the roots of infected SNF plants was not significantly increased, suggesting no large change in JA level in roots (electronic supplementary material, figure S5). Therefore, the role of JA on the observed effect on nodules has to be clarified. An interesting point is that the JA and SA-signalling pathways are known to interact antagonistically in dicotyledonous plants but that both of them seemed to be activated in SNF plants. Although activation may have occurred at different time points during the 12 days of infestation, the JA and SA effects were still visible at the end of the infestation period. More work is required to test if and how the presence of *S. meliloti* modifies the JA/SA interplay in case of stress.

## 5. Conclusion

Sowing and inoculating plant seeds with rhizobia is a method to improve the growth of plants by helping them to adapt to poor nitrogen conditions, to improve soil fertility and limit the use of chemical fertilizers and thus the greenhouse effect. Beneficial soil microbes can also improve the plant defences against pathogens and insect herbivores through the ISR.

Our data reveal an interplay between rhizobia and aphid infestation through the modulation of plant growth, nitrogen fixation symbiosis and defence responses. *Rhizobium* symbiosis did not protect *Medicago* from aphid infestation, but significantly reduced aphid fitness compared to NI plants. In return, aphid infestation decreased the number of root nodules and nitrogen fixation in growing SNF plants, thereby reducing the benefit of symbiosis, and therefore the interest of legumes for nitrogen enrichment of the soil. In a context of a more widespread use of legumes, this study shows that plants in symbiosis and without symbiosis may interact with pests in a very different way. The generality of our results yet remains to be tested by considering the genetic diversity of the different partners. Indeed, both plant and aphid genotypes may influence the outcome of the interaction [70]. This is also true for the rhizobia for which other strains should be used under the same conditions to compare the effect of aphids according to the efficiency of the symbiotic interaction.

Data accessibility. Statistical data used are provided in the electronic supplementary material and the dataset are available from the Dryad Digital Repository: https://doi.org/10.5061/dryad.q2bvq83gt [71].

Authors' contributions. M.P., P.F., J.-L.G.: supervision and funding acquisition; M.P., P.F., J.-L.G and G.P.: experimental design and conceptualization; G.P.: data acquisition and original draft preparation; G.P. and P.F.: statistical analysis; J.-C.S.: supply of aphid biological resources; G.P., J.-C.S., J.-L.G., M.P. and P.F: writing-review and editing.

Competing interests. The authors declare no conflict of interest.

Funding. Financial support was provided by the University Côte d'Azur (UCA) and the Department of Plant Health (SPE) from the French National Institute for Research in Agriculture, Food and Environment (INRAE). The project was also supported by the LABEX SIGNALIFE 'Investments for the future' ANR-11-LABX-0028 (http://signalife.unice.fr). G.P. was funded by the LABEX SIGNALIFE from UCA.

Acknowledgements. We are grateful to the symbiosis team members for their help and assistance in the experimental work. We also acknowledge Dr S. Tares for her help with aphid rearing and the ESIM team members for constant help and support.

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
