## [Reviewer comments · Proceedings of the Royal Society B: Biological Sciences]

Review History

RSPB-2020-0346.R0 (Original submission)

Review form: Reviewer 1

Recommendation

Major revision is needed (please make suggestions in comments)

Scientific importance: Is the manuscript an original and important contribution to its field?

Good

General interest: Is the paper of sufficient general interest?

Good

Quality of the paper: Is the overall quality of the paper suitable?

Excellent

Is the length of the paper justified?

Yes

Should the paper be seen by a specialist statistical reviewer?

No

Do you have any concerns about statistical analyses in this paper? If so, please specify them explicitly in your report.

No

It is a condition of publication that authors make their supporting data, code and materials available - either as supplementary material or hosted in an external repository. Please rate, if applicable, the supporting data on the following criteria.

Is it accessible?

Yes

Is it clear?

Yes

Is it adequate?

Yes

Do you have any ethical concerns with this paper?

Yes

Comments to the Author

The paper deals with a novel and scientifically sound question regarding the sensitivity of *Medicago truncatula* in symbiosis with Rhizobacteria to pests, such as the pea aphid. They found that aphid weight was reduced when feeding on Rhizobium-inoculated plants. However, Rhizobium symbiosis do not protect *Medicago* plants from pea aphids, which were able to reduce root nodules and reduce nitrogen fixation. The paper has merit, is well written but the way that aphid fitness was determined is not well explained nor well justified.

Some further comments:

Abstract:

Line 36. Authors did not study the effect of the symbiotic state (Rhizobium inoculated vs. non-inoculated) on aphid growth, but its effect on aphid weight. It is not the same.

Introduction:

Line 62: Aphids do not consume plant phloem, they compete with plants for assimilates (soluble CH) present in the phloem.

Line 80: The sentence: "This while the symbiotic state of the plant could impact the phenotypes observed in aphids". Do you mean that plants in symbiosis with Rhizobacteria may have an impact on the phenotypes observed in aphids ? Please re-write the sentence-

Lines 89-90 and thereafter: The way that the Rhizobium-inoculated and non-inoculated plants are labeled is not very intuitive. I suggest to replace the label of NFS to RI, and keep the non-inoculated as NI. Or alternatively, the inoculated plants could be label simply as I and non-inoculated as NI.

Methods:

Lines 108-109: The following sentence is not clear "After germination, 6 seedlings were transferred per pots". Do you mean that after germination 6 seedlings were transferred to each pot ?

Line 131 and thereafter: How was aphid fitness assessed? It is clear that survival was recorded daily and aphid weight on day 12, but there is no explanation on the methods used to assess aphid fitness. What was the aphid age when starting the fitness experiment ? Where all aphids of the same age when the experiments started ? Was aphid fecundity assessed ? Was the

developmental time determined? Why was aphid weight recorded on day 12? No details at all on how aphid fitness was determined.

It is well known that aphid fitness is related with aphid weight (eg Dixon AFG (1987) Parthenogenetic Reproduction and the Rate of increase in Aphid, Vol. 2A: Aphids. Their biology, natural enemies and control. (ed. by AK Minks & P Harrewijn), pp. 269-285.). However, authors fail to explain why aphid weight in day 12 is used instead of determining other parameters such as the intrinsic rate of natural increase, r_m .

Results: I would recommend not to use the word "performance" when referring to aphid weight (eg as illustrated in Figure 1 or line 180). Aphid performance needs to be determined by recording aphid fecundity and development, which can give an accurate indication of growth rate. What the authors have determined is the weight of aphids, but not their performance.

Discussion:

Lines 244-245. Authors suggest that they studied the effect of the nitrogen source of legumes on the growth rate of the pea aphid. That is not true, they only studied the effect of nitrogen source on aphid weight. The same mistake is repeated on lines 248 and thereafter.

Review form: Reviewer 2

Recommendation

Major revision is needed (please make suggestions in comments)

Scientific importance: Is the manuscript an original and important contribution to its field?

Good

General interest: Is the paper of sufficient general interest?

Excellent

Quality of the paper: Is the overall quality of the paper suitable?

Good

Is the length of the paper justified?

Yes

Should the paper be seen by a specialist statistical reviewer?

No

Do you have any concerns about statistical analyses in this paper? If so, please specify them explicitly in your report.

No

It is a condition of publication that authors make their supporting data, code and materials available - either as supplementary material or hosted in an external repository. Please rate, if applicable, the supporting data on the following criteria.

Is it accessible?

N/A

Is it clear?

N/A

Is it adequate?

N/A

Do you have any ethical concerns with this paper?

No

Comments to the Author

The MS by Pandharikar et al aims at the study of interactions in a complex system that includes two regimes of N supply to the plants (symbiotic or nitrate), and several lines of aphids carrying different types of facultative bacterial symbionts. They conclude that the nitrogen status affects the growth of aphids, and that the presence of aphids reduces plant growth irrespective of the nitrogen status. They also describe an effect of the specific facultative aphid symbiont on the performance of the nodulated plants. The potential contribution of facultative symbionts is quite interesting, although it could be better exploited here.

The data are clearly presented, and in general the conclusions are supported by the data. I have some concerns:

The data shown in Fig. 2 show that nitrate-fed plants accumulate about two-fold of dry matter as compared to nodulated plants. This indicates a bad symbiotic performance in the control condition. The authors should comment on this on the discussion taking into account the limited symbiotic effectiveness described for *M truncatula* A17 in association with the sister strain *S. meliloti* 1021 derived from the same isolate as Sm1021 (Terpolilli et al, 2008; Kacmierczak et al., 2017 MPMI). Poorly N-fed plants could be compromised to cope with aphid infection, and this might affect the results with the different facultative symbionts.

The figure 3 is not very clear. They claim a 20% difference between Control and Amp treatments in Fig 3C, which is not evident from the histogram. Maybe a Table could show better these data.

Also in this topic, it is interesting that the both number and weight of nodules are reduced, when they are linked to reduced plant dry weight (and likely to reduced nitrogen fixed). This suggests the possibility that there is a real blockage of nodulation as a consequence of aphid infestation, and the plant does not compensate by generating more nodules. The hypothesis they consider of different allocation of resources is appealing, and it might have an hormone-based mechanism. The JA hormone, either alone or by its ratio to ethylene has been linked to blocking of nodulation (Sun et al, 2006 Plant J). The authors should comment on this possibility (see comment and reference below on JA effect on nodulation)

Do the authors consider that, according to what is observed in Fig 3B and D, the presence of FS might in part compensate for the deleterious effect of aphid infestation? Reduction is higher in lines without FS, and with Ss symbiont.

The data shown in Fig. 3E indicate that leghemoglobin expression is too variable as to be considered a reliable marker. The authors should try a different nodulin gene (GS/GOGAT or other) to assess symbiotic performance on the plant side.

In my opinion, the discussion leaves out several relevant points. In the first topic there is not a real discussion on the data, but just a summary of them. It is clear that the type of N nutrition affects the growth of the aphids. The authors should discuss whether this could be due to a different composition of the phloem sap, or to the defence response of the plant. Is it known whether this composition is different according to N source? Some references are available for the composition of *M truncatula*, (Sandstrom & Peterson, 1994; Sulieman et al., 2010).

Provided there is an effect of the aphid facultative symbiont on the legume symbiosis, it is expected that such effect should be different depending on the facultative symbiont. Have the authors any idea on the mechanism for the observed differences associated to the presence of

different FS?

The authors state in the discussion (line 327) that it is not known whether JA acts on nodulation. There is at least one reference not considered (Sun et al., 2006, *Plant J* 46:961) showing an effect of this hormone in nodule formation.

In the Conclusion section the authors state that the damages caused by the aphid infestation is “a counterproductive effect in using legumes ...” It should be considered that the symbiotic combination used is far from optimal in terms of N feeding to the plant. In addition, the same effect is seen with nitrate-fed plants, and probably with other species, so I don't fully understand why to link the effect to symbiotic nitrogen fixation.

Expression of MtCp6 causes nodule senescence. Is it possible that aphid attack induces CP6 expression, and this causes the senescence?

Minor points.

The title sounds a bit weird “Nitrogen-fixation symbiosis...” Is not correct. Maybe: “Symbiotic nitrogen fixation...” or something similar.

Lines 212-213: the meaning of the sentences is a bit contradictory: there is a non significant difference with YR2-Ri(a) in specific ARA, that becomes “no more significant”?

Line 341. Rhizobium symbiosis DOES NOT protect Medicago from APHID infestation

Line 163 Calculs?

Decision letter (RSPB-2020-0346.R0)

20-Mar-2020

Dear Dr Gatti:

I am writing to inform you that your manuscript RSPB-2020-0346 entitled "Nitrogen-fixation symbiosis affects legume-aphid interactions and plant defence" has, in its current form, been rejected for publication in *Proceedings B*.

This action has been taken on the advice of referees, who have recommended that substantial revisions are necessary. With this in mind we would be happy to consider a resubmission, provided the comments of the referees are fully addressed. However please note that this is not a provisional acceptance.

Sincerely,
 Dr Sasha Dall
 mailto: proceedingsb@royalsociety.org

Associate Editor
 Board Member: 1

Comments to Author:

Two expert have now reviewer your article "Nitrogen-fixation symbiosis affects legume-aphid interactions and plant defence" (RSPB-2020-0346). Both consider that the study is novel, relevant and of general interest. Both of them, however, point to major problems in the manuscript.

Reviewer 1 points that, contrary to claims in the text, aphid performance is not quantified, but just aphid weigh 12 days post infestation. Performance should have been better estimated from aphid development time or/and fecundity. Why was aphid weight used, and how were aphid experiments performed need to be clarified.

Reviewer 2 addresses the main question that the symbiotic combination sued is far from optimal in terms of the plant N nutrition, which may affect all the results of the paper. Related to this point he7he also advises that a gene other than leghemoglobin is used to monitor symbiosis on the plant side.

Both reviewers also point to many necessary modifications in the text, related to the interpretation of the results and to the precision in the use of concepts, which I also consider of importance.

Reviewer(s)' Comments to Author:

Referee: 1

Comments to the Author(s)

The paper deals with a novel and scientifically sound question regarding the sensitivity of *Medicago truncatula* in symbiosis with Rhizobacteria to pests, such as the pea aphid. They found that aphid weight was reduced when feeding on Rhizobium-inoculated plants. However, Rhizobium simbiosis do not protect *Medicago* plants from pea aphids, which were able to reduce root nodules and reduce nitrogen fixation. The paper has merit, is well written but the way that aphid fitness was determined is not well explained nor well justified.

Some further comments:

Abstract:

Line 36. Authors did not study the effect of the symbiotic state (Rhizobium inoculated vs. non-inoculated) on aphid growth, but its effect on aphid weight. It is not the same.

Introduction:

Line 62: Aphids do not consume plant phloem, they compete with plants for assimilates (soluble CH) present in the phloem.

Line 80: The sentence: "This while the symbiotic state of the plant could impact the phenotypes observed in aphids". Do you mean that plants in symbiosis with Rhizobacteria may have an impact on the phenotypes observed in aphids? Please re-write the sentence-

Lines 89-90 and thereafter: The way that the Rhizobium-inoculated and non-inoculated plants are labeled is not very intuitive. I suggest to replace the label of NFS to RI, and keep the non-inoculated as NI. Or alternatively, the inoculated plants could be label simply as I and non-inoculated as NI.

Methods:

Lines 108-109: The following sentence is not clear "After germination, 6 seedlings were transferred per pots". Do you mean that after germination 6 seedlings were transferred to each pot?

Line 131 and thereafter: How was aphid fitness assessed? It is clear that survival was recorded daily and aphid weight on day 12, but there is no explanation on the methods used to assess aphid fitness. What was the aphid age when starting the fitness experiment? Were all aphids of the same age when the experiments started? Was aphid fecundity assessed? Was the developmental time determined? Why was aphid weight recorded on day 12? No details at all on how aphid fitness was determined.

It is well known that aphid fitness is related with aphid weight (eg Dixon AFG (1987) Parthenogenetic Reproduction and the Rate of increase in Aphid, Vol. 2A: Aphids. Their biology, natural enemies and control. (ed. by AK Minks & P Harrewijn), pp. 269-285.). However, authors fail to explain why aphid weight in day 12 is used instead of determining other parameters such as the intrinsic rate of natural increase, r_m .

Results: I would recommend not to use the word "performance" when referring to aphid weight (eg as illustrated in Figure 1 or line 180). Aphid performance needs to be determined by recording aphid fecundity and development, which can give an accurate indication of growth rate. What the authors have determined is the weight of aphids, but not their performance.

Discussion:

Lines 244-245. Authors suggest that they studied the effect of the nitrogen source of legumes on the growth rate of the pea aphid. That is not true, they only studied the effect of nitrogen source on aphid weight. The same mistake is repeated on lines 248 and thereafter.

Referee: 2

Comments to the Author(s)

The MS by Pandharikar et al aims at the study of interactions in a complex system that includes two regimes of N supply to the plants (symbiotic or nitrate), and several lines of aphids carrying different types of facultative bacterial symbionts. They conclude that the nitrogen status affects the growth of aphids, and that the presence of aphids reduces plant growth irrespective of the nitrogen status. They also describe an effect of the specific facultative aphid symbiont on the performance of the nodulated plants. The potential contribution of facultative symbionts is quite interesting, although it could be better exploited here.

The data are clearly presented, and in general the conclusions are supported by the data. I have some concerns:

The data shown in Fig. 2 show that nitrate-fed plants accumulate about two-fold of dry matter as compared to nodulated plants. This indicates a bad symbiotic performance in the control condition. The authors should comment on this on the discussion taking into account the limited symbiotic effectiveness described for *M. truncatula* A17 in association with the sister strain *S. meliloti* 1021 derived from the same isolate as Sm1021 (Terpolilli et al, 2008; Kacmierczak et al., 2017 MPMI). Poorly N-fed plants could be compromised to cope with aphid infection, and this might affect the results with the different facultative symbionts.

The figure 3 is not very clear. They claim a 20% difference between Control and Amp treatments in Fig 3C, which is not evident from the histogram. Maybe a Table could show better these data.

Also in this topic, it is interesting that the both number and weight of nodules are reduced, when they are linked to reduced plant dry weight (and likely to reduced nitrogen fixed). This suggests the possibility that there is a real blockage of nodulation as a consequence of aphid infestation, and the plant does not compensate by generating more nodules. The hypothesis they consider of different allocation of resources is appealing, and it might have an hormone-based mechanism. The JA hormone, either alone or by its ratio to ethylene has been linked to blocking of nodulation (Sun et al, 2006 Plant J). The authors should comment on this possibility (see comment and reference below on JA effect on nodulation)

Do the authors consider that, according to what is observed in Fig 3B and D, the presence of FS might in part compensate for the deleterious effect of aphid infestation? Reduction is higher in lines without FS, and with Ss symbiont.

The data shown in Fig. 3E indicate that leghemoglobin expression is too variable as to be considered a reliable marker. The authors should try a different nodulin gene (GS/GOGAT or other) to assess symbiotic performance on the plant side.

In my opinion, the discussion leaves out several relevant points. In the first topic there is not a real discussion on the data, but just a summary of them. It is clear that the type of N nutrition affects the growth of the aphids. The authors should discuss whether this could be due to a different composition of the phloem sap, or to the defence response of the plant. Is it known whether this composition is different according to N source? Some references are available for the composition of *M truncatula*, (Sandstrom & Peterson, 1994; Sulieman et al., 2010).

Provided there is an effect of the aphid facultative symbiont on the legume symbiosis, it is expected that such effect should be different depending on the facultative symbiont. Have the authors any idea on the mechanism for the observed differences associated to the presence of different FS?

The authors state in the discussion (line 327) that it is not known whether JA acts on nodulation. There is at least one reference not considered (Sun et al., 2006, Plant J 46:961) showing an effect of this hormone in nodule formation.

In the Conclusion section the authors state that the damages caused by the aphid infestation is “a counterproductive effect in using legumes ...” It should be considered that the symbiotic combination used is far from optimal in terms of N feeding to the plant. In addition, the same effect is seen with nitrate-fed plants, and probably with other species, so I don’t fully understand why to link the effect to symbiotic nitrogen fixation.

Expression of MtCp6 causes nodule senescence. Is it possible that aphid attack induces CP6 expression, and this causes the senescence?

Minor points.

The title sounds a bit weird “Nitrogen-fixation symbiosis...” Is not correct. Maybe: “Symbiotic nitrogen fixation...” or something similar.

Lines 212-213: the meaning of the sentences is a bit contradictory: there is a non significant difference with YR2-Ri(a) in specific ARA, that becomes “no more significant”?

Line 341. Rhizobium symbiosis DOES NOT protect Medicago from APHID infestation

Line 163 Calculs?

Author's Response to Decision Letter for (RSPB-2020-0346.R0)

See Appendix A.

RSPB-2020-1493.R0

Review form: Reviewer 1

Recommendation

Accept with minor revision (please list in comments)

Scientific importance: Is the manuscript an original and important contribution to its field?

Good

General interest: Is the paper of sufficient general interest?

Good

Quality of the paper: Is the overall quality of the paper suitable?

Good

Is the length of the paper justified?

Yes

Should the paper be seen by a specialist statistical reviewer?

No

Do you have any concerns about statistical analyses in this paper? If so, please specify them explicitly in your report.

No

It is a condition of publication that authors make their supporting data, code and materials available - either as supplementary material or hosted in an external repository. Please rate, if applicable, the supporting data on the following criteria.

Is it accessible?

N/A

Is it clear?

N/A

Is it adequate?

N/A

Do you have any ethical concerns with this paper?

No

Comments to the Author

Most of my comments have been solved, but there is still a problem with one of the paragraphs in the new version of the paper:

Rewrote line 62-64: "Among the 4,000 known aphid species, 450 thrive on crops and about 200 cause serious damage by sucking up the plant phloem, reducing plant growth, and, most importantly, transmitting plant viruses"

Aphids do not suck up the plant phloem, what they really do is to suck up the phloem sap. The phloem is a complex of cell tissues that contain parenchyma, companion cells and sieve elements. Only the sieve elements contain phloem sap, which is what the aphids ingest after stylet insertion.

Therefore, you should change the sentence accordingly.

Review form: Reviewer 2

Recommendation

Accept with minor revision (please list in comments)

Scientific importance: Is the manuscript an original and important contribution to its field?

Good

General interest: Is the paper of sufficient general interest?

Excellent

Quality of the paper: Is the overall quality of the paper suitable?

Excellent

Is the length of the paper justified?

Yes

Should the paper be seen by a specialist statistical reviewer?

No

Do you have any concerns about statistical analyses in this paper? If so, please specify them explicitly in your report.

No

It is a condition of publication that authors make their supporting data, code and materials available - either as supplementary material or hosted in an external repository. Please rate, if applicable, the supporting data on the following criteria.

Is it accessible?

N/A

Is it clear?

N/A

Is it adequate?

N/A

Do you have any ethical concerns with this paper?

No

Comments to the Author

My comments on the author's answers:

Question 1: The answer of the authors only partially solves the question asked. They agree on that the specific interaction studied (*M. truncatula* A17/ *S. meliloti* 2011) is not optimal, and they include a sentence on that on the revised version of the manuscript, but they support it with a reference in which this problem has a different focus, as they do not compare it with other situations. In my view, the reference of Terpolilli et al (2008) is more appropriate, since these authors show that it is not the symbiosis by itself that is less efficient (as supported by Moreau et al), but the particular strain, as compared to other strains. I am not asking the authors to repeat the experiments with a rhizobial strain more suited to *M. truncatula*. It would be enough to include the idea that a deficient nitrogen nutrition, described for this strain but not for other strains, could be partly responsible for a worse response of the nodulated plant to aphid infection. A 50% reduction of plant biomass (cited in line 253), is essentially the level of reduction shown by Terpolilli et al for this particular symbiotic association. Other strains, however, go up to less than 20% reduction, so the SNF by itself is not necessarily that costly for plant growth when compared to nitrate as source.

Question 2. There was in fact a mistake in my question, since the 20% difference cited was between strains other than the ones I mentioned. I agree with the correction introduced (but not with the comment included in the response stating that there is 20% decrease in data on fig. 3C. Table S11 shows that Amp treatment represents a 13.3% decrease vs control)

Question 3. I agree with the addition of the reference, but the introduced sentence is not completely true. There could be an effect linked to JA, since at least one of the FS strains lead to a highly significant increase in P1 expression, and the mean expression of P1 of the other strains are much higher than the control. It looks like it is a problem of variability of the results that makes it not significant. The authors should describe better what they show in Fig. S4.

Question 5. FS-containing aphids lead to a stronger reduction in MtLb expression. It is still surprising that such big differences in MtLb (in Ss and Ri(n)) are not significant regarding the control plant. On the other hand, the results based on MtLb do not fit with the effect of FS on plant dry weight, nodule mass, and particularly ARA per plant, and is contradictory with what is argued in the response to Question 4. With the correction introduced, do the authors conclude that FS are responsible for the reduction in nitrogen fixation, or for offsetting the reduction due to aphids?

Question 6. The authors have not included any comment on the composition of the phloem sap under nitrate fed/SNF situations, and not a single reference on sap composition of any type of plant has been included. The sentences included in lines 281-290 do not cover this point, but the effect of aphids, which is a different thing. There could be a basal difference between SNF/NI. Maybe it is not published, but they can certainly refer to this possibility to explain their results.

Question 7. The authors include in their response some interesting references pertinent to what it was commented, but I am only one of the many potential readers of the paper. Why not including some of it on the manuscript?

Question 8. I still think that the general conclusion should be toned down. The fact that an inefficient strain is being used could have relevance for the effect of aphids on legumes. To conclude something so general about nitrogen fixation the data should be obtained with an efficient legume/rhizobium combination.

Questions 9-10: I agree with responses provided by authors to other minor points.

Question 11: Regarding the title, the need to correct the first words somehow impaired me to see the rest of it in the previous version. I do not think it is appropriate to state that ... Nitrogen fixing symbiosis affects ... plant defence, since there is no difference on the expression of plant

defence genes when aphids are not present (line 224 and Fig 4). It is the other way around, as said in the last subtitle of the Discussion section: the aphid infestation has a different effect on NI and SNF plants. So the title should have the aphid infection as subject, not the nitrogen fixation.

Question 12-13: I agree with the changes

Decision letter (RSPB-2020-1493.R0)

20-Jul-2020

Dear Dr Gatti:

Your manuscript has now been peer reviewed and the reviews have been assessed by an Associate Editor. The reviewers' comments (not including confidential comments to the Editor) and the comments from the Associate Editor are included at the end of this email for your reference. As you will see, the reviewers and the Editors have raised some concerns with your manuscript and we would like to invite you to revise your manuscript to address them.

Research ethics:

Use of animals and field studies:

It is a condition of publication that you make available the data and research materials supporting the results in the article (<https://royalsociety.org/journals/authors/author-guidelines/#data>). Datasets should be deposited in an appropriate publicly available repository and details of the associated accession number, link or DOI to the datasets must be included in the Data Accessibility section of the article (<https://royalsociety.org/journals/ethics-policies/data-sharing-mining/>). Reference(s) to datasets should also be included in the reference list of the article with DOIs (where available).

Please submit a copy of your revised paper within three weeks. If we do not hear from you within this time your manuscript will be rejected. If you are unable to meet this deadline please let us know as soon as possible, as we may be able to grant a short extension.

Best wishes,

Dr Sasha Dall

Associate Editor

Comments to Author:

Your manuscript "Symbiotic nitrogen fixation affects legume-aphid interactions and plant defence" (RSPB-2020-1493) has now been reviewed by the same two experts who reviewed the initial submission. While Reviewer 1 is satisfied with how his/her concerns were handled, he/she still point to an imprecise sentence that should be mended.

Reviewer 2 has more concerns, but they affect the text, and should be easily addressed.

Reviewer(s)' Comments to Author:

Referee: 2

Comments to the Author(s).

My comments on the author's answers:

Question 1: The answer of the authors only partially solves the question asked. They agree on that the specific interaction studied (*M. truncatula* A17/ *S. meliloti* 2011) is not optimal, and they include a sentence on that on the revised version of the manuscript, but they support it with a reference in which this problem has a different focus, as they do not compare it with other situations. In my view, the reference of Terpolilli et al (2008) is more appropriate, since these authors show that it is not the symbiosis by itself that is less efficient (as supported by Moreau et al), but the particular strain, as compared to other strains. I am not asking the authors to repeat the experiments with a rhizobial strain more suited to *M. truncatula*. It would be enough to include the idea that a deficient nitrogen nutrition, described for this strain but not for other strains, could be partly responsible for a worse response of the nodulated plant to aphid infection. A 50% reduction of plant biomass (cited in line 253), is essentially the level of reduction shown by Terpolilli et al for this particular symbiotic association. Other strains, however, go up to less than 20% reduction, so the SNF by itself is not necessarily that costly for plant growth when compared to nitrate as source.

Question 2. There was in fact a mistake in my question, since the 20% difference cited was between strains other than the ones I mentioned. I agree with the correction introduced (but not with the comment included in the response stating that there is 20% decrease in data on fig. 3C. Table S11 shows that Amp treatment represents a 13.3% decrease vs control)

Question 3. I agree with the addition of the reference, but the introduced sentence is not completely true. There could be an effect linked to JA, since at least one of the FS strains lead to a highly significant increase in P1 expression, and the mean expression of P1 of the other strains are much higher than the control. It looks like it is a problem of variability of the results that makes it not significant. The authors should describe better what they show in Fig. S4.

Question 5. FS-containing aphids lead to a stronger reduction in MtLb expression. It is still surprising that such big differences in MtLb (in Ss and Ri(n)) are not significant regarding the control plant. On the other hand, the results based on MtLb do not fit with the effect of FS on plant dry weight, nodule mass, and particularly ARA per plant, and is contradictory with what is argued in the response to Question 4. With the correction introduced, do the authors conclude that FS are responsible for the reduction in nitrogen fixation, or for offsetting the reduction due to aphids?

Question 6. The authors have not included any comment on the composition of the phloem sap under nitrate fed/SNF situations, and not a single reference on sap composition of any type of plant has been included. The sentences included in lines 281-290 do not cover this point, but the effect of aphids, which is a different thing. There could be a basal difference between SNF/NI. Maybe it is not published, but they can certainly refer to this possibility to explain their results.

Question 7. The authors include in their response some interesting references pertinent to what it was commented, but I am only one of the many potential readers of the paper. Why not including some of it on the manuscript?

Question 8. I still think that the general conclusion should be toned down. The fact that an inefficient strain is being used could have relevance for the effect of aphids on legumes. To conclude something so general about nitrogen fixation the data should be obtained with an efficient legume/rhizobium combination.

Questions 9-10: I agree with responses provided by authors to other minor points.

Question 11: Regarding the title, the need to correct the first words somehow impaired me to see the rest of it in the previous version. I do not think it is appropriate to state that ... Nitrogen fixing symbiosis affects ... plant defence, since there is no difference on the expression of plant defence genes when aphids are not present (line 224 and Fig 4). It is the other way around, as said in the last subtitle of the Discussion section: the aphid infestation has a different effect on NI and SNF plants. So the title should have the aphid infection as subject, not the nitrogen fixation.

Question 12-13: I agree with the changes

Referee: 1

Comments to the Author(s).

Most of my comments have been solved, but there is still a problem with one of the paragraphs in the new version of the paper:

Rewrote line 62-64: "Among the 4,000 known aphid species, 450 thrive on crops and about 200 cause serious damage by sucking up the plant phloem, reducing plant growth, and, most importantly, transmitting plant viruses"

Aphids do not suck up the plant phloem, what they really do is to suck up the phloem sap. The phloem is a complex of cell tissues that contain parenchyma, companion cells and sieve elements. Only the sieve elements contain phloem sap, which is what the aphids ingest after stylet insertion.

Therefore, you should change the sentence accordingly.

Author's Response to Decision Letter for (RSPB-2020-1493.R0)

See Appendix B.

RSPB-2020-1493.R1 (Revision)

Review form: Reviewer 1

Recommendation

Accept as is

Scientific importance: Is the manuscript an original and important contribution to its field?

Good

General interest: Is the paper of sufficient general interest?

Excellent

Quality of the paper: Is the overall quality of the paper suitable?

Excellent

Is the length of the paper justified?

Yes

Should the paper be seen by a specialist statistical reviewer?

No

Do you have any concerns about statistical analyses in this paper? If so, please specify them explicitly in your report.

No

It is a condition of publication that authors make their supporting data, code and materials available - either as supplementary material or hosted in an external repository. Please rate, if applicable, the supporting data on the following criteria.

Is it accessible?

N/A

Is it clear?

N/A

Is it adequate?

N/A

Do you have any ethical concerns with this paper?

No

Comments to the Author

No further comments.

Decision letter (RSPB-2020-1493.R1)

07-Aug-2020

Dear Dr Gatti

I am pleased to inform you that your manuscript entitled "Aphid infestation differently affects the defences of nitrate-fed and nitrogen-fixing *Medicago truncatula* and alters symbiotic nitrogen fixation." has been accepted for publication in *Proceedings B*.

Open Access

Paper charges

Sincerely,

Dr Sasha Dall

Appendix A

From: Pr. Pierre Frendo and M. Poirié
Sophia Agrobiotech Institute, Sophia Antipolis, France

To: Dr Sasha Dall,
Editor of Proceedings of the Royal Society B

Sophia Antipolis,

Dear editor,

Thank you for considering the revised version of our manuscript (**RSPB-2020-0346**) "**Symbiotic nitrogen fixation affects legume-aphid interactions and plant defense**" for publication in the *Proceedings of the Royal Society of London B*. We are grateful to the reviewers and the Associate Editor for their assistance and we appreciated that they found our work very interesting, the data to be of good quality and contributing to field interactions between plants and aphids. As you summarized in your letter, the reviewers highlighted two main areas that could be improved and made a number of useful comments. The manuscript has now been revised in light of these advices and you will find our specific responses to all of the comments detailed below. We have also lightly edited the text to try to shorten and improved it.

We very much hope that you will find our revised manuscript suitable for publication in Proceedings B and we are looking forward to receiving your answer.

Yours sincerely,

Responses to reviewers

Reviewer #1

We thank the reviewer for insightful comments and helpful suggestions.

“...the way that aphid fitness was determined is not well explained nor well justified.”

1) Abstract: Line 36. Authors did not study the effect of the symbiotic state (*Rhizobium* inoculated vs. non-inoculated) on aphid growth, but its effect on aphid weight. It is not the same.

2) Discussion: Lines 244-245. Authors suggest that they studied the effect of the nitrogen source of legumes on the growth rate of the pea aphid. That is not true, they only studied the effect of nitrogen source on aphid weight. The same mistake is repeated on lines 248 and thereafter.

We agree with the reviewer comments. Therefore, we replaced “the growth of aphids” by “the weight of aphids” in the abstract and result section. However, as mentioned by the reviewer itself “*It is well known that aphid fitness is related with aphid weight (e.g. Dixon AFG (1987) Parthenogenetic Reproduction and the Rate of increase in Aphid, Vol. 2A: Aphids. Their biology, natural enemies and control. (ed. by AK Minks & P Harrewijn), pp. 269-285.)*”. We therefore kept the term “fitness” in the discussion since weight and survival are indeed considered as good fitness proxies, and we added the proposed reference (reference [45]).

3) Introduction: Line 62: Aphids do not consume plant phloem; they compete with plants for assimilates (soluble CH) present in the phloem.

We agree with the reviewer comment although many publications have used the word “consume” to refer to “drinking phloem sap” in the context of aphid damage (The Arabidopsis Book Arabidopsis thaliana—Aphid Interaction First published on May 22, 2012: e0159. doi: 10.1199/tab.0159. Review article. “Plant defence against aphids, the pest extraordinaire” <https://doi.org/10.1016/j.plantsci.2018.04.027>). That said, since the word “consume” may lead to a false interpretation, we replaced it with “sucking up”.

Rewrote line 62-64: “Among the 4,000 known aphid species, 450 thrive on crops and about 200 cause serious damage by sucking up the plant phloem, reducing plant growth, and, most importantly, transmitting plant viruses”.

4) Line 80: The sentence: "This while the symbiotic state of the plant could impact the phenotypes observed in aphids". Do you mean that plants in symbiosis with Rhizobacteria may have an impact on the phenotypes observed in aphids? Please re-write the sentence.

We thank the reviewer for pointing out the complexity in the sentence. We therefore rewrote it. Line 81: “However, the nitrogen-fixing symbiosis of the plant could in turn affect the aphid phenotypes”.

5) Lines 89-90 and thereafter: The way that the Rhizobium-inoculated and non-inoculated plants are labelled is not very intuitive. I suggest replacing the label of NFS to RI and keep the non-inoculated as NI. Or alternatively, the inoculated plants could be label simply as I and non-inoculated as NI.

We understand the suggestion, but the proposed alternatives for Rhizobium-inoculated plants are problematic. For “RI”, we used a line of aphids hosting *Regiella insecticola* which is labeled “Ri”, which could possibly be confusing for readers. For the letter “I” alone, it could be mistaken for the I of the first person or even be considered a typo.

We therefore decided to call the Rhizobium-inoculated plants, “SNF” for “Symbiotic Nitrogen Fixation”, which is widely used in the literature and fits with the change in the title. We also believe that this will facilitate the distinction between the two types of plants for the reader.

6) Methods: Lines 108-109: The following sentence is not clear "After germination, 6 seedlings were transferred per pots". Do you mean that after germination 6 seedlings were transferred to each pot?

This is exactly what we did. We agree that it was not clear enough, so we rewrote the sentence. *Rewrote line 108-111 (Mat and Meth)*

7) Line 131 and thereafter: How was aphid fitness assessed? It is clear that survival was recorded daily and aphid weight on day 12, but there is no explanation on the methods used to assess aphid fitness. What was the aphid age when starting the fitness experiment? Where all aphids of the same age when the experiments started? Was aphid fecundity assessed? Was the developmental time determined? Why was aphid weight recorded on day 12? No details at all on how aphid fitness was determined.However, authors fail to explain why aphid weight in day 12 is used instead of determining other parameters such as the intrinsic rate of natural increase, r_m .

As already stated above, the weight and survival of aphids are considered proxies for their fitness. The link between the weight of aphids and fecundity has notably been widely demonstrated.

Regarding fertility, we followed in a preliminary study the reproduction of aphids up to 12 days after their removal from the plant, on isolated bean leaves. Although we did not exactly quantify the reproduction rate, we observed that the offspring number for all lines of aphids feeding on NI plants was higher than those on SNF plants, except for *Ri(a)*. The result for *Ri(a)* may be due to the high mortality rate in the transferred mothers during the observation period. Thus, a greater weight of aphids in our conditions also seems to be well correlated with their fertility. This has been added *in the discussion line 242-245* as “Adult females of all lines were also able to reproduce during at least two weeks on SNF or NI plants and preliminary data suggest a higher offspring number on NI plants than on SNF, except for YR2-*Ri(a)*, and therefore a positive correlation between weight and fecundity (Gaurav Pandharikar, personal observation).”.

We did not use the intrinsic rate (r_m) because it would have required monitoring the production of offspring and we quickly had too many samples to follow in our protocol. Furthermore, the experimental design was not set-up for this (for example, the plants used could not feed a large number of aphids for several weeks, so we had to move the aphids on fava bean leaves, a different support) and the removing of newborn aphids would have require a large number of manipulations of the plant and aphids which was not possible.

Aphid age is indicated line 128 mat et meth (“Ten synchronized L1 nymphs per pot were used for infestation”). All the details on the production of aphids and plants and the experiments are provided in supplementary files (**Figure S1**).

The weight of the aphids was recorded on day 12 because this period of time, in our protocol, allowed the development of aphids from the L1 stage (the stage at which they were put on the plant) until the adult stage but before they started reproducing (we also did not want to

manipulate the aphids to weigh them every day). We also expected 12 days to be sufficient time for the establishment of the nitrogen-fixing symbiosis and the plant's defense response to aphids, allowing the analysis of associated parameters. This explanation is added in the manuscript lines 131-133.

8) Results: I would recommend not to use the word "performance" when referring to aphid weight (e.g. as illustrated in Figure 1 or line 180). Aphid performance needs to be determined by recording aphid fecundity and development, which can give an accurate indication of growth rate. What the authors have determined is the weight of aphids, but not their performance.

We agree with the reviewer that using the word “performance” could lead to false interpretation since the fertility of aphids was not evaluated.

To be more accurate, we rewrote the text line 182 in “Effect of SNF and NI Plants on the survival and weight of pea aphid lines”, and the title of Figure 1 as “Survival and weight of pea aphids of the different lines on *M. truncatula* SNF and NI plants”.

Reviewer #2

1) The data shown in Fig. 2 show that nitrate-fed plants accumulate about two-fold of dry matter as compared to nodulated plants. This indicates a bad symbiotic performance in the control condition. The authors should comment on this on the discussion taking into account the limited symbiotic effectiveness described for *M. truncatula* A17 in association with the sister strain *S. meliloti* 1021 derived from the same isolate as Sm1021 (Terpolilli et al, 2008; Kacmierczak et al., 2017 MPMI). Poorly N-fed plants could be compromised to cope with aphid infection, and this might affect the results with the different facultative symbionts.

We understand the reviewer comment. The association between *M. truncatula* A17 and *S. meliloti* strains 2011 or 1021 is conventionally used in experiments on the nitrogen fixing symbiosis whereas the interactions between *M. truncatula* and other strains of rhizobium such as *R. medicae* are much less described. We therefore agree that the analysis of the effect of the genetic diversity of rhizobium on aphid infection is an interesting scientific question but beyond the scope of our manuscript.

The growth retardation of SNF plants certainly results from the metabolic difference of nitrogen with NI plants. However, the symbiotic fixation of nitrogen costs more energy than the supply of nitrate. This certainly partly explains the regulation by shoots of the symbiotic nitrogen fixation and the senescence of root nodules when nitrogen-fixing plants are treated with nitrate. This difference between the SNF and the nitrate feeding was analyzed by Moreau and colleagues (2008) (J. Exp. Bot. 59, 3509-3522 (doi:10.1093/jxb/ern203)) who showed that this ‘model symbiotic association’ does not allow the plant to meet its nitrogen requirements when nitrogen fixation is the main nitrogen source for plant growth.

We nevertheless agree that the association of *M. truncatula* A17 with the bacterium isolate Sm2011 might be involved in the retardation of SNF plants due to limited symbiotic effectiveness. This has been added in the discussion lines 256.

It is also true that the presence or absence of N compounds in the plant sieve may or may not be favorable to certain aphid lines due to the help of their endosymbionts. However, although differences have been observed between the aphid lines, they have almost always added to a general effect of aphids on plants and are, therefore, rather difficult to assess clearly.

2) *The figure 3 is not very clear. They claim a 20% difference between Control and Amp treatments in Fig 3C, which is not evident from the histogram. Maybe a Table could show better these data.*

We agree with the reviewer that figure 3C may be difficult to analyse. We actually mentioned a significant reduction in nitrogen fixation activity in plants infested with Amp, Hd and Ss lines compared to the control, and it is true that a difference of 20% could be calculated for Amp from the graph. We also reported a non-significant 20% difference between the control and YR2-*Ri*(n) and YR2-*Ri*(a). We realized since then that it was a typing mistake as it was rather a non-significant reduction of approximately 10%. We have kept Figure 3 as it was in the previous text and we have added the corresponding data table in the supplementary material (Table S11) given the space limitation.

3) *Also, in this topic, it is interesting that the both number and weight of nodules are reduced, when they are linked to reduced plant dry weight (and likely to reduced nitrogen fixed). This suggests the possibility that there is a real blockage of nodulation as a consequence of aphid infestation, and the plant does not compensate by generating more nodules. The hypothesis they consider of different allocation of resources is appealing, and it might have a hormone-based mechanism. The JA hormone, either alone or by its ratio to ethylene has been linked to blocking of nodulation (Sun et al, 2006 Plant J). The authors should comment on this possibility (see comment and reference below on JA effect on nodulation).*

4) *The authors state in the discussion (line 327) that it is not known whether JA acts on nodulation. There is at least one reference not considered (Sun et al., 2006, Plan J 46:961) showing an effect of this hormone in nodule formation.*

This is a very important point raised by the reviewer. However, for information, we tested the expression of *PI* in roots and found no significant induction in our experiments, which does not support a direct local JA-mediated suppression of nodulation.

We have introduced the reference Sun et al. 2006 and these data (as a supplementary Figure S4) in the manuscript line 319-320. Such a JA-mediated response in shoots could nevertheless possibly inhibit nodulation in *M. truncatula* plants.

4) *Do the authors consider that, according to what is observed in Fig 3B and D, the presence of FS might in part compensate for the deleterious effect of aphid infestation? Reduction is higher in lines without FS, and with Ss symbiont.*

We observed that the presence of certain FS has a negative impact on aphids, restricting their ability to feed on SNF plants, and therefore lowering the induced reduction of nitrogen fixation. We have indeed observed that the aphid lines Amp (without FS) and Ss (hosting *Serratia*) affected plant biomass and nitrogen fixation levels more than those hosting the other endosymbionts. This might actually suggest that *Ri* and *Hd* could somehow offset the effect of

the aphid infestation. However, as this is the first study into how the aphid and host plant symbionts modulate their interaction, more data need to be produced to draw a firm conclusion on the differential effects of FS. In the future, we plan to address into more details how a particular endosymbiotic composition of aphids could modulate plant-aphid interactions.

5) The data shown in Fig. 3E indicate that leghemoglobin expression is too variable as to be considered a reliable marker. The authors should try a different nodulin gene (GS/GOGAT or other) to assess symbiotic performance on the plant side.

Leghemoglobin is reported to be a very good marker for nodule function (Msehli et al., 2019, doi: 10.1002/jpln.201800233, Li et al., 2020 doi:10.3389/fpls.2020.00137). We agree that it was more difficult to reach a clear conclusion under our conditions as we observed a high standard error in the control plants. This was actually due to outliers in the control genes qPCR (housekeeping genes) used to calibrate *leg* expression. Analysis of qPCR data without these outliers reduced the SE in *leg* expression in the control plants and we observed a statistically significant reduction in *leg* expression in *Ri(a)* and *Hd*, and very close to the significance for *Ri(n)*. Both previous and new results are provided below.

The amplitude of the repression of *MtLb1* agrees with the small reduction in ARA per mg of nodule observed (only about 20%). Thus, in this context, the absence of a strong alteration in the expression of leghemoglobin was not surprising. We therefore changed the Figure and the Table using the recalculated data without outliers in the revised manuscript.

Old Figure

Old Table

Sample	MtLb1	SEM	t-test
Control	1	0.429236	
Amp	0.840018	0.419561	0.780022
Ri(n)	0.275871	0.117741	0.141657
Ri(a)	0.399391	0.121624	0.251172
Hd	0.311871	0.066601	0.134981
Ss	0.255556	0.115614	0.174654

New Figure

New Table

Sample	MtLb1	SEM	t-test
Control	1	0.294353	
Amp	0.746941	0.419838	0.436704
Ri(n)	0.235613	0.091887	0.055844
Ri(a)	0.377525	0.067156	0.041158
Hd	0.340977	0.060217	0.034613
Ss	0.212046	0.09498	0.128093

6) In my opinion, the discussion leaves out several relevant points. In the first topic there is not a real discussion on the data, but just a summary of them. It is clear that the type of N nutrition affects the growth of the aphids. The authors should discuss whether this could be due to a different composition of the phloem sap, or to the defense response of the plant. Is it known whether this composition is different according to N source? Some references are available for the composition of *M truncatula*, (Sandstrom & Peterson, 1994; Sulieman et al., 2010).

To our knowledge, the composition of *M. truncatula* phloem sap has been analyzed (Sandström & Pettersson, 1994; Sulieman et al., 2010) but not according to the N source, such analyses being indeed scarce. Song et al., (2017) showed that *Medicago sativa* plants with Rhizobia accumulate more antioxidants, osmolytes, organic acids and metabolites involved in nitrogen fixation. Therefore, phloem analysis is important not only to describe metabolites but also to identify possible increases in secondary metabolites in the sap which could act as defense molecules against aphids.

We have added sentences in the discussion part to refer to these two elements (line 281-290). “One hypothesis would be the occurrence of a trade-off in the plant, the aphid infestation leading to the inhibition of the costly formation of nodules to compensate for the uptake of metabolites from the sieve. This would in turn decrease the availability of nitrogen-containing metabolites, such as amino acids, for aphids, which can decrease their appetite for the plant. Another way, but not exclusive, by which aphid infestation may affect the function of nodules is through the activation of plant defence pathways”.

7) Provided there is an effect of the aphid facultative symbiont on the legume symbiosis, it is expected that such effect should be different depending on the facultative symbiont. Have the authors any idea on the mechanism for the observed differences associated to the presence of different FS?

We have not studied the mechanism involved in this process (to our knowledge, data on the effect of FS on aphid metabolism are scarce, and mainly under stress), but we observed certain effects which seemed to be specific to aphid FS.

Frago et al., 2017 (DOI: 10.1038/s41467-017-01935-0) have shown that the presence of aphid endosymbionts modulates the systemic release of volatile substances by plants after an aphid attack, increasing their fitness. Likewise, we believe that the presence (of some) or absence of aphid endosymbionts could possibly modulate plant-aphid interactions.

Alternatively, the variation in the weight of aphids may be due to the cost of the multiplication of symbionts during the development of aphids (Doremus and Oliver, 2017) in competition with that of *Buchnera* (Simonet et al., 2016) and aphids for nutrients.

Buchnera density was positively correlated with aphid dietary nitrogen levels, while that of the facultative symbiont *Serratia symbiotica* increased in aphids reared on a low-nitrogen diet, indicating possibly distinct regulatory mechanisms or nutritional needs between compulsory and optional symbionts in the same host insect (Wilkinson et al., 2007). Deciphering the effect of plant symbiosis on the multiplication and function of different symbionts via metabolomic studies could be an interesting future development of this work.

8) In the Conclusion section the authors state that the damages caused by the aphid infestation is “a counterproductive effect in using legumes ...” It should be considered that the symbiotic combination used is far from optimal in terms of N feeding to the plant. In addition, the same effect is seen with nitrate-fed plants, and probably with other species, so I don’t fully understand why to link the effect to symbiotic nitrogen fixation.

The use of legumes is particularly promoted because of their symbiotic capacity for fixing nitrogen, limiting the use of nitrogen fertilizers. If the infestation by aphids in particular tends to reduce this capacity of fixation and promote the senescence of the nodules, the expected effect of soil enrichment will be reduced and therefore somehow the interest of using legumes in the field.

We agree that the sentence was unclear and we rewrote it line 335-338 as " *In return, aphid infestation decreased the root nodules' number and nitrogen fixation in SNF plants, thereby reducing the benefit of symbiosis and therefore the interest of legumes for nitrogen enrichment of the soil*".

9) Line 341. *Rhizobium* symbiosis DOES NOT protect Medicago from APHID infestation

We agree with the reviewer. To avoid misinterpretation, we have rewritten this part of the conclusion, in line 353-354 as "***Rhizobium* symbiosis did not protect Medicago from aphid infestation, but significantly reduced aphid fitness compared to NI plants**".

10) Expression of MtCp6 causes nodule senescence. Is it possible that aphid attack induces CP6 expression, and this causes the senescence?

Studies have identified the *MtCp6* gene as a marker of the stress-induced nodule senescence (Pierre et al., 2014). However, with our present knowledge, it cannot be identified whether the induction of the *MtCp6* gene following an aphid infestation is direct or indirect.

11) The title sounds a bit weird “Nitrogen-fixation symbiosis...” Is not correct. Maybe: “Symbiotic nitrogen fixation...” or something similar.

Thanks for the suggestion, we changed the title to “*Symbiotic nitrogen fixation affects legume-aphid interactions and plant defence*”.

12) Lines 212-213: the meaning of the sentences is a bit contradictory: there is a non-significant difference with YR2-Ri(a) in specific ARA, that becomes “no more significant”?

Thanks for reporting the contradictory sentence, we rewrote this sentence in line 208-209 as “*The ARA per plant gave a similar result, except for the reduction induced by YR2-Ri(a) which was significant here*”.

13) Line 163 *Calculs*? Thanks for reporting the mistake, we changed it to calculations (line 165).

Responses to Referees

Referee: 1

Comments to the Author(s).

Most of my comments have been solved, but there is still a problem with one of the paragraphs in the new version of the paper:

Rewrote line 62-64: "Among the 4,000 known aphid species, 450 thrive on crops and about 200 cause serious damage by sucking up the plant phloem, reducing plant growth, and, most importantly, transmitting plant viruses". Aphids do not suck up the plant phloem, what they really do is to suck up the phloem sap. The phloem is a complex of cell tissues that contain parenchyma, companion cells and sieve elements. Only the sieve elements contain phloem sap, which is what the aphids ingest after stylet insertion. Therefore, you should change the sentence accordingly.

We have changed the sentence to clarify that aphids suck plant phloem sap (line 62-63).

Referee: 2

Comments to the Author(s).

My comments on the author's answers:

*Question 1: The answer of the authors only partially solves the question asked. They agree on that the specific interaction studied (*M. truncatula* A17/ *S. meliloti* 2011) is not optimal, and they include a sentence on that on the revised version of the manuscript, but they support it with a reference in which this problem has a different focus, as they do not compare it with other situations. In my view, the reference of Terpolilli et al (2008) is more appropriate, since these authors show that is not the symbiosis by itself that is less efficient (as supported by Moreau et al), but the particular strain, as compared to other strains. I am not asking the authors to repeat the experiments with a rhizobial strain more suited to *M. truncatula*. It would be enough to include the idea that a deficient nitrogen nutrition, described for this strain but not for other strains, could be partly responsible for a worse response of the nodulated plant to aphid infection. A 50% reduction of plant biomass (cited in line 253), is essentially the level of reduction shown by Terpolilli et al for this particular symbiotic association. Other strains, however, go up to less than 20% reduction, so the SNF by itself is not necessarily that costly for plant growth when compared to nitrate as source.*

We have introduced in the manuscript the reference of Terpolilli (2008) (ref 55) indicating that similarly *M. truncatula* A17 / *S. meliloti* 2011 are not the most efficient pair (line 257). We have also added a sentence suggesting that the association with this rhizobial strain could be partly responsible for the lower plant weight and weak nodulation and therefore for the strong effect of aphid infestation on nodulation (line 257-259).

Question 2. There was in fact a mistake in my question, since the 20% difference cited was between strains other than the ones I mentioned. I agree with the correction introduced (but not with the

comment included in the response stating that there is 20% decrease in data on fig. 3C. Table S11 shows that Amp treatment represents a 13.3% decrease vs control).

Thank you for pointing out the mistake in our response. We agree with your response that Amp treatment represents a 13.3% decrease vs control.

Question 3. I agree with the addition of the reference, but the introduced sentence is not completely true. There could be an effect linked to JA, since at least one of the FS strains lead to a highly significant increase in PI expression, and the mean expression of PI of the other strains are much higher than the control. It looks like it is a problem of variability of the results that makes it not significant. The authors should describe better what they show in Fig. S4.

We agree with the reviewer that JA may be involved in the nodulation phenotype observed upon infestation although this is not noticeable with most of the aphid lines (non-significant difference in PI expression). However, further experiments will be required to test this hypothesis. We have now added a sentence to better describe the results of Figure S4 as suggested by the reviewer (line 228-229).

Question 5. FS-containing aphids lead to a stronger reduction in MtLb expression. It is still surprising that such big differences in MtLb (in Ss and Ri(n)) are not significant regarding the control plant. On the other hand, the results based on MtLb do not fit with the effect of FS on plant dry weight, nodule mass, and particularly ARA per plant, and are contradictory with what is argued in the response to Question 4. With the correction introduced, do the authors conclude that FS are responsible for the reduction in nitrogen fixation, or for offsetting the reduction due to aphids?

We have rechecked the statistical analysis of our experiment and observed that the lack of significance resulted very likely from the variability observed in the control plants.

We do not fully agree that the expression of *MtLb* does not fit with the other nodulation parameters. This depends on the aphid lines used and therefore on the symbionts hosted. *MtLb* relative gene expression is roughly correlated to the ARA per mg of nodule (nitrogen fixation efficiency, fig3C) and this parameter is not strongly affected by aphid infestation. The ARA per plant takes into account the efficiency of nodulation (number of nodules per plant, fig3A), the weight of the nodule in mg per plant (development of the nodule, fig3B) and the efficiency of nitrogen fixation per mg of nodule (fig3C).

We do not believe that a clear conclusion can be drawn about the effect of aphid facultative symbionts (FS) on reducing nitrogen fixation since the aphid line without FS reduces the efficiency of nodulation and there is a large variation in *MtLb* expression in plants infested with this line. It would therefore be difficult to conclude that FS are responsible for the reduction in nitrogen fixation. That said, it remains plausible that some of the FS, here *R. insecticola* and *H. defensa*, might somehow offset the reduction due to aphid infestation, as discussed from line 288. And we fully agree that more experiments are required to disentangle the effect of the different FS.

Question 6. The authors have not included any comment on the composition of the phloem sap under nitrate fed/SNF situations, and not a single reference on sap composition of any type of plant has been included. The sentences included in lines 281-290 do not cover this point, but the effect of aphids, which is a different thing. There could be a basal difference between SNF/NI. Maybe it is not published, but they can certainly refer to this possibility to explain their results.

We have introduced a sentence on the potential effect of the composition of the sap that may differ between SNF and NI plants (line 296-299). There are indeed very few metabolomic studies comparing the sap of legumes with and without symbiosis and especially none at the stage of plant growth we used.

Question 7. The authors include in their response some interesting references pertinent to what it was commented, but I am only one of the many potential readers of the paper. Why not including some of it on the manuscript?

Thank you for your suggestion. We would have liked to include more references in the manuscript, but it already contains a large number and we are limited in terms of the number of words. We therefore had to decide on few. We added more explanation one from line 235-237 referenced to Frago et al. 2017 and second from line 292-296 referenced to Wilkinson et al. 2007.

Question 8. I still think that the general conclusion should be toned down. The fact that an inefficient strain is being used could have relevance for the effect of aphids on legumes. To conclude something so general about nitrogen fixation the data should be obtained with an efficient legume/rhizobium combination.

In the conclusion, we had indicated that the generality of our results had to be tested with other plant and aphid genotypes (line 350-352). We also mention now that other strains of rhizobia should be used under the same conditions to compare the effect of aphids according to the efficiency of the symbiotic interaction (line 352-354).

Question 11: Regarding the title, the need to correct the first words somehow impaired me to see the rest of it in the previous version. I do not think it is appropriate to state that ... Nitrogen fixing symbiosis affects ... plant defence, since there is no difference on the expression of plant defence genes when aphids are not present (line 224 and Fig 4). It is the other way around, as said in the last subtitle of the Discussion section: the aphid infestation has a different effect on NI and SNF plants. So, the title should have the aphid infection as subject, not the nitrogen fixation.

We have changed the title to reflect the reviewer's comment and we have also included the effect of aphids on the nitrogen fixation. The new title is “**Aphid infestation differently affects the defences of nitrate-fed and nitrogen-fixing *Medicago truncatula* and alters symbiotic nitrogen fixation**”.

Questions 9-10: I agree with responses provided by authors to other minor points.

Question 12-13: I agree with the changes.

That's fine for us, no further change needed.